# Neural Description Logic Reasoning over Incomplete Knowledge Bases

## Abstract

Concept learning exploits background knowledge in the form of description logic axioms to learn explainable classification models from knowledge bases. Despite recent breakthroughs in the runtime of concept learners, most approaches still cannot be deployed on real-world knowledge bases. This is due to their use of description logic reasoners, which do not scale to large datasets. Moreover, these reasoners are not robust against inconsistencies and erroneous data, both being hallmarks of real datasets. We address this challenge by presenting a novel neural reasoner dubbed EBR. Our reasoner relies on embeddings to rapidly approximate the results of a symbolic reasoner. We show that EBR solely requires retrieving instances for atomic concepts and existential restrictions to retrieve the instances of any concept in $\mathcal{SROIQ}$. Importantly, our experiments also suggest that EBR is robust against missing and erroneous data.

## 1 Introduction

Description Logics (DLs) (Baader, 2003) offer a formal framework for structured knowledge representation and reasoning. Due to their well-defined semantics and favorable computational properties, DLs have become essential tools in fields such as ontology engineering (Keet, 2018), knowledge representation (Brachman & Levesque, 2004), and semantic web technologies (Horrocks et al., 2003).

Logical entailment is one of the most extensively studied reasoning mechanisms in computer science (Tang et al., 2022). It is also a crucial task in the exploration of DL Knowledge Bases (KBs). Formally, a statement $\varphi$ is logically entailed by a KB if $\varphi$ is true in every model of the KB (Tang et al., 2022). In DL KBs, such entailments are typically computed by reasoners (termed as symbolic reasoners) including Pellet (Sirin et al., 2007), Fact++ (Tsarkov & Horrocks, 2006), HermiT (Glimm et al., 2014), and RacerPro (Haarslev et al., 2012). These reasoners are sound and complete; the statements they derive are correct, and they derive every entailed statement.

Although symbolic reasoners are being successfully applied to infer missing knowledge on benchmark datasets, their application at a large scale has been hindered by their inability to handle inconsistencies, inferring missing instance assertions, and their impractical runtimes. An inconsistent KB logically entails every statement trivially. To illustrate this, let $\mathcal{K} = (\{C \sqcap D \sqsubseteq A, B \sqsubseteq \bot\}, \{C(a), D(a), B(b)\})$ be a KB. In this case, a classical symbolic reasoner cannot determine the membership of the individual $a$ in $A$ although it is not involved in any inconsistency. Another issue that arises is incompleteness; illustrated via the following example. Let $\mathcal{K} = (\{\texttt{Person}(Bob), \texttt{Person}(Paul), \texttt{Person}(Ani), \texttt{knows}(Bob, Paul), \texttt{knows}(Ani, Joe)\})$. In $\mathcal{K}$, a symbolic reasoner cannot infer the membership for $Joe$ in the class $\texttt{Person}$. Indeed, the issues of inconsistency and incompleteness pose significant challenges, as most large-scale KBs, such as Wikidata, DBpedia, and Yago, are often incomplete or inconsistent (Töpper et al., 2012; Nickel et al., 2015; Krompaß et al., 2015). Furthermore, the aforementioned state-of-the-art reasoners operate on a single CPU, which hinders scalability to large real-world datasets by not leveraging modern parallel computing architectures.

Neural link predictors have been extensively investigated to deal with incompleteness on various datasets (Dettmers et al., 2018; Ren & Leskovec, 2020). The likelihood of assertion (e.g. a class membership $\texttt{Person}(?)$ or $\texttt{knows}(Ani, ?)$) can be computed through learning continuous vector representations elucidated in Section 2.2. Recent works showed that neural link predictors can be

effectively applied to answer complex queries involving multi-model reasoning (Arakelyan et al., 2021; van Krieken et al., 2022; Bai et al., 2023; Demir et al., 2023; Arakelyan et al., 2024).

Our approach dubbed EBR (**E**mbedding **B**ased **R**easoner) leverages knowledge graph embeddings to perform reasoning over incomplete and inconsistent knowledge bases. We employ a neural link predictor to facilitate the retrieval of missing data and handle inconsistencies. Our contributions can be summarized as follows:

- We propose neural semantics to tackle the instance retrieval problem on incomplete or inconsistent $\mathcal{SROIQ}$ KBs.

- We provide an in-depth comparison for instance retrieval against symbolic reasoners on six datasets (Father, Family, Semantic Bible, Mutagenesis, Carcinogenesis, and Vicodi). We show that on knowledge bases with varying numbers of missing assertions, our approach outperforms symbolic approaches, which often return an empty result set in this case.

- We show that the instance retrieval problem can be tackled without storing knowledge bases in memory. Storing the learned parameters of a neural link predictor suffices to retrieve the instances of any $\mathcal{SROIQ}$ concepts. Importantly, the inference time can be decreased by leveraging GPUs, enabling efficient handling of large-scale computations.

## 2 BACKGROUND AND RELATED WORKS

A DL KB $\mathcal{K}$ consists of a TBox $\mathcal{T}$ and an ABox $\mathcal{A}$, where the former specifies the schema (i.e., the axioms that describe the structure of the domain being modelled) and the latter contains the data (i.e., the assertions describing the objects in a domain of discourse). Precisely, a TBox contains general concept inclusions (GCIs) of the form $C \sqsubseteq D$, where $C, D$ are concepts. Moreover, the ABox includes assertions having the form $C(a)$ (concept assertion) or $r(a, b)$ (role assertion), for individuals $a, b$, concept $C$, and role $r$. The syntax and semantics for concepts in $\mathcal{SROIQ}$ Baader (2003); Hitzler et al. (2009) are given in Table 1.

Table 1: Syntax & semantics for $\mathcal{SROIQ}$ concepts. $\mathcal{I}$ stands for an interpretation with domain $\Delta^{\mathcal{I}}$.

| Construct | Syntax | Semantics |
|---|---|---|
| Atomic concept | $A$ | $A^{\mathcal{I}} \subseteq \Delta^{\mathcal{I}}$ |
| Role | $r$ | $r^{\mathcal{I}} \subseteq \Delta^{\mathcal{I}} \times \Delta^{\mathcal{I}}$ |
| Top concept | $\top$ | $\Delta^{\mathcal{I}}$ |
| Bottom concept | $\bot$ | $\emptyset$ |
| Negation | $\neg C$ | $\Delta^{\mathcal{I}} \setminus C^{\mathcal{I}}$ |
| Conjunction | $C \sqcap D$ | $C^{\mathcal{I}} \cap D^{\mathcal{I}}$ |
| Disjunction | $C \sqcup D$ | $C^{\mathcal{I}} \cup D^{\mathcal{I}}$ |
| Existential restriction | $\exists\, r.C$ | $\{x \mid \exists\, y.(x, y) \in r^{\mathcal{I}} \wedge y \in C^{\mathcal{I}}\}$ |
| Universal restriction | $\forall\, r.C$ | $\{x \mid \forall\, y.(x, y) \in r^{\mathcal{I}} \implies y \in C^{\mathcal{I}}\}$ |
| Universal Role | $U$ | $\Delta^{\mathcal{I}} \times \Delta^{\mathcal{I}}$ |
| Inverse Role | $r^{-1}$ | $\{(y, x) \mid (x, y) \in r^{\mathcal{I}}\}$ |
| Nominals | $\{o\}$ | $\{o\}^{\mathcal{I}} \subseteq \Delta^{\mathcal{I}}$ |
| At least restriction | $\geq n\, r.C$ | $\{a \mid\ |\{b \in C | (a, b) \in r^{\mathcal{I}}\}| \geq n\}$ |
| At most restriction | $\leq n\, r.C$ | $\{a \mid\ |\{b \in C | (a, b) \in r^{\mathcal{I}}\}| \leq n\}$ |

Let $C \sqsubseteq D$ be a GCI and $\mathcal{I}$ be an interpretation. Then, $\mathcal{I}$ satisfies $C \sqsubseteq D$, denoted as $\mathcal{I} \models C \sqsubseteq D$ iff $C^{\mathcal{I}} \sqsubseteq D^{\mathcal{I}}$. Similarly, $\mathcal{I}$ satisfies an assertion $C(a)$ iff $a^{\mathcal{I}} \in C(a)$; and the assertion $r(a, b)$ iff $(a^{\mathcal{I}}, b^{\mathcal{I}}) \in r^{\mathcal{I}}$. We write an axiom to mean either a TBox GCI or an ABox assertion. We say that $\mathcal{I}$ is a model of the KB $\mathcal{K}$, denoted by $\mathcal{I} \models \mathcal{K}$, iff $\mathcal{I}$ satisfies every axiom in $\mathcal{K}$. Finally, let $\mathcal{K}$ be a DL KB and $\alpha$ be an axiom, then $\mathcal{K} \models \alpha$ iff $\mathcal{I} \models \alpha$ for every model $\mathcal{I}$ of $\mathcal{K}$. The DL $\mathcal{SROIQ}$ additionally admits an RBox $\mathcal{R}$ which includes (1) a *role hierarchy* $\mathcal{R}_h$ consisting of (generalised) role inclusion axioms of the form $R \sqsubseteq S$, and (2) a set $\mathcal{R}_a$ of *role assertions* stating, for instance, that a role $R$ must be reflexive/irreflexive, symmetric/asymmetric, transitive, and that two roles $R$ and $S$ are disjoint. The semantics for RBox axioms is defined analogously Horrocks et al. (2006).

Reasoning with expressive DL KBs is a computationally hard task. Specifically, the instance checking problem, which, given $\mathcal{K}$, a concept $C$, and an individual $x$, determines whether $x$ is an instance of $C$ in $\mathcal{K}$ (denoted as $\mathcal{K} \models C(x)$), has a high computational complexity. For the DL $\mathcal{SROIQ}$, this problem is non-deterministic double exponential time complete (Kazakov, 2008). Given such high complexity and the additional challenges posed by incomplete and inconsistent data in real-world scenarios, practical applications often require the use of approximation algorithms.

In the following, we first briefly introduce symbolic DL reasoners suitable for small, consistent, and complete datasets. Next, we introduce knowledge graph embeddings, able to deal with incompleteness and inconsistency issues, albeit for simple 1-hop queries. We continue with neural query answering approaches that generalize the capabilities of knowledge graph embeddings, e.g., supporting multi-hop queries with conjunctions and disjunctions. Finally, we provide an overview of approaches supporting expressive description logics including reasoning with type hierarchies.

## 2.1 Symbolic Reasoning over Knowledge Bases

To the best of our knowledge, HermiT (Glimm et al., 2014) is the only reasoner that fully supports the OWL 2 standard, including all the datatypes specified in the standard and correctly reasons about properties as well as classes. It is based on a novel "hypertableau" calculus that addresses performance problems due to nondeterminism and model size—the primary sources of complexity in state-of-the-art OWL reasoners. HermiT reduces all basic reasoning tasks, including subsumption test, to satisfiability checking. HermiT was shown to outperform the previous reasoners Pellet (Sirin et al., 2007), and Fact++ (Tsarkov & Horrocks, 2006). Similarly, OWL2Bench (Singh et al., 2020) compares different reasoners across datasets and OWL profiles. Pellet and its extension Openllet were found to perform best in terms of runtime. JFact, the Java implementation of Fact++, performed worst on all the reasoning tasks across all OWL 2 profiles. While OWL2Bench assessed the reasoners performance to detect inconsistent ontologies, they did not assess the robustness of reasoners, i.e., how well they answer queries on incomplete/inconsistent data.

## 2.2 Knowledge Graph Embeddings

A plethora of Knowledge Graph Embedding (KGE)/neural link predictors models have been developed over the last decade (Dettmers et al., 2018). Most KGE models learn continuous vector representations tailored towards link prediction. They are often defined as parameterized scoring functions $\phi_\Theta : \mathcal{E} \times \mathcal{R} \times \mathcal{E} \mapsto \mathbb{R}$, where $\Theta$ denotes parameters and often comprise entity embeddings $\mathbf{E} \in \mathbb{R}^{|\mathcal{E}| \times d_e}$, relation embeddings $\mathbf{R} \in \mathbb{R}^{|\mathcal{R}| \times d_r}$, and additional parameters (e.g., affine transformations, batch normalizations, convolutions). Since $d_e = d_r$ holds for many state-of-the-art models, we will use $d$ to signify the number of real parameters used for the embedding of an entity or relation. Given $(\mathtt{h}, \mathtt{r}, \mathtt{t}) \in \mathcal{E} \times \mathcal{R} \times \mathcal{E}$, the prediction $\hat{y} := \phi_\Theta(\mathtt{h}, \mathtt{r}, \mathtt{t})$ signals the likelihood of $(\mathtt{h}, \mathtt{r}, \mathtt{t})$ being true. Since $\mathcal{G}$ contains only assertions that are assumed to be true, assertions assumed to be false are often generated by applying the negative sampling, 1vsAll or Kvsall training strategies (Ruffinelli et al., 2020). KGE have been successfully applied to link prediction (Dai et al., 2020; Wang et al., 2021), drug discovery (Bonner et al., 2022), community detection (Hamilton et al., 2017), question answering (Hamilton et al., 2018), and product recommendation (Choudhary et al., 2021).

## 2.3 Neural Query Answering on Incomplete Knowledge Graphs

In recent years, significant progress has been made on querying incomplete triple-based Knowledge Graphs (KGs) that are represented as subject-predicate-object triples, such as those in RDF. Hamilton et al. (2018) laid the foundations for multi-hop reasoning with graph query embeddings (GQE). Given a conjunctive query, they learn continuous vector representations for queries, entities, and relations and answer queries by performing projection and intersection operations in the embedding vector space. Ren et al. (2020) show that GQE cannot answer Existential Positive First-order (EPFO) queries since GQE does not model the union operator. Hence, they propose Query2Box that represents an EPFO query with a set of box embeddings, where one box embedding is constructed per conjunctive subquery. A query is answered by returning the entities whose minimal distance to one of the box embeddings is smallest. TeMP (Hu et al., 2022) builds on top of GQE and allows each entity to have a set of types.

All the aforementioned models learn query embeddings and answer queries via nearest neighbor search in the embedding space. However, learning embeddings for complex, multi-hop queries involving conjunctions and disjunctions can be computationally demanding. Towards this end, Arakelyan et al. (2021) propose complex query decomposition (CQD). They answer EPFO queries by decomposing them into single-hop subqueries and aggregate the scores of a pre-trained single-hop link predictor (e.g., ComplEx-N3). Scores are aggregated using a t-norm and t-conorm—continuous generalizations of the logical conjunction and disjunction (Arakelyan et al., 2021; Klement et al., 2004). Their experiments suggest that CQD outperforms GQE and Query2Box; it generalizes well to complex query structures while requiring orders of magnitude less training data. Zhu et al. (2022) highlight that CQD is the only interpretable model among the aforementioned models as it produces intermediate results. Recently, Demir et al. (2023) extended CQD to answer multi-hop queries involving literals. Andresel et al. (2023) extend both GQE and CQD to answer queries in the presence of an ontolgoy. They do so via query rewriting and ontology-aware knowledge graph embeddings. However, the expressiveness of their queries is limited. Unlike our approach, they only support Existential Positive First-Order (EPFO) queries, but do not support negations, universal restrictions, and cardinality restrictions.

### 2.4 DESCRIPTION LOGICS EMBEDDINGS

The previously discussed neural reasoning approaches assume a triple-based data model for knowledge graphs consisting of subject-predicate-object triples. Recently, there has been a growing interest in generating vector representations (embeddings) for OWL ontologies. However, most of them do not allow answering instance queries, i.e., retrieve all instances in a given concept.

Several embedding techniques have been proposed for the lightweight DLs including $\mathcal{EL}$ and $\mathcal{EL}^{++}$ (Kulmanov et al., 2019; Xiong et al., 2022; Peng et al., 2022; Lacerda et al., 2023; Jackermeier et al., 2024). The underlying idea involves representing concepts as geometrical shapes (boxes or balls). Further, Mondal et al. (2021) proposed to map the concepts and roles in an ontology to $n$-dimensional vector, and Singh et al. (2021) proposed a reinforcement learning-based solution, both targeting the subsumption task in $\mathcal{EL}$ ontologies. However, these methods do not support the construction of complex axioms involving negation or disjunction, nor do they support instance retrieval.

For more expressive DLs such as $\mathcal{ALC}$, embedding techniques have been proposed (Özçep et al., 2020; Hohenecker & Lukasiewicz, 2020; Tang et al., 2022; Zhapa-Camacho & Hoehndorf, 2023a;b; Özcep et al., 2023) with a primary focus on representing an ontology geometrically. The main motivation of these approaches lies in proving that an ontology is satisfiable if it admits a satisfying *geometrical* structure. For $\mathcal{SROIQ}$, Holter et al. (2019) and Chen et al. (2021) proposed mapping OWL ontologies to RDF graphs and applying Word2Vec (Church, 2017) over generated walks, considering concept membership and subsumption tasks. For an overview of existing works and their techniques, we refer the reader to the survey by Chen et al. (2024).

Finally, Hohenecker & Lukasiewicz (2020) proposed a neural architecture to perform logical entailment over Datalog rules. Their approach considers instance checking (the entailment of instance queries) in the context of *data complexity*. That is, the TBox in the KB is fixed, whereas the input includes a query and a fixed-size ABox. Thus, their approach (1) is suitable in a setting where the TBox remains fixed, and (2) addresses only specific aspects of reasoning within an ontological framework.

## 3 METHODOLOGY

The methodology of EBR consists of three main components: the embedding model, prediction mechanisms, and the mapping of DL syntax to a neural semantic syntax. The source code of EBR is provided in the supplemental material.

### 3.1 EMBEDDING MODEL

As we will see in Section 3.3, mapping description logic syntax to our neural semantics requires only an engine that can answer queries of the form $(x, \texttt{rdf:type}, ?)$, $(x, r, ?)$, $(?, r, y)$, and $(x, ?, y)$.

Therefore, we first extract assertions and axioms of the form $C(a) \equiv (x, \texttt{rdf:type}, C)$, $r(x, y) \equiv (x, r, y)$, and $C \sqsubseteq D \equiv (C, \texttt{rdfs:subClassOf}, D)$ from a given knowledge base to construct a knowledge graph $\mathcal{G} \subseteq \mathcal{E} \times \mathcal{R} \times \mathcal{E}$. We then use a KGE model to learn embeddings for entities and relation types in the constructed graph. This yields a trained KGE model $\phi_\Theta : \mathcal{E} \times \mathcal{R} \times \mathcal{E} \to \mathbb{V}^d$ ($\mathbb{V}^d$ is a vector space) which can answer the aforementioned queries. In our experiments, we employ the state-of-the-art model KECI (Demir & Ngonga Ngomo, 2023) (with $p = 0$, $q = 1$) for embedding computation. In these settings, KECI is equivalent to ComplEx (Trouillon et al., 2016), which embeds entities and relation types into a complex vector space. Therefore, unless stated elsewhere, $\phi_\Theta$ is defined as

$$\phi_\Theta : \mathcal{E} \times \mathcal{R} \times \mathcal{E} \to \mathbb{C}^d; \ \phi_\Theta(\mathbf{x}, \mathbf{r}, \mathbf{y}) = \mathcal{R}e(\langle \mathbf{x}, \mathbf{r}, \bar{\mathbf{y}} \rangle).$$

Here, $\mathbf{x}$, $\mathbf{r}$, and $\mathbf{y}$ are complex embeddings of the head entity $x$, the relation type $r$, and the tail entity $y$, respectively; $\overline{\mathbf{y}}$ denotes the complex conjugate of $\mathbf{y}$.

## 3.2 PREDICTION MECHANISM

Once the model $\phi_\Theta$ is trained, we construct a neural link predictor $\phi : \mathcal{E} \cup \mathcal{R} \mapsto [0, 1]$ which can assign a score to every triple $(x, r, y)$ based on the provided query. For a given query with missing tail $(\texttt{x}, \texttt{r}, ?)$ or missing head $(?, \texttt{r}, \texttt{y})$, we rank all possible entities (individuals or atomic concepts) $x \in \mathcal{E}$ or $y \in \mathcal{E}$ based on the score $\phi(\texttt{x}, \texttt{r}, \texttt{y})$. Higher scores indicate more likely matches. The same technique applies if there is a missing relation. That is, given a query $(\texttt{x}, ?, \texttt{y})$, we rank all possible relation types $r \in \mathcal{R}$ based on the score $\phi(\texttt{x}, \texttt{r}, \texttt{y})$ and higher scores indicate potential matches. Therefore, for any triple $(\texttt{x}, \texttt{r}, \texttt{y})$ in the knowledge graph representation of a knowledge base, the score of the triple is computed as $\text{score}(\texttt{x}, \texttt{r}, \texttt{y}) = \phi(\texttt{x}, \texttt{r}, \texttt{y})$.

## 3.3 MAPPING DL SYNTAX TO NEURAL SEMANTICS

The syntax and semantics for concepts in $\mathcal{SROIQ}$ are provided in the appendix. We define a mapping from DL semantics to neural semantics to bridge the gap between DLs and neural embeddings.

- **Atomic concept.** The embedding-based retrieval (EBR) of an atomic concept $A$ is defined as the set of individuals $x$ for which the link predictor $\phi$ returns a score greater than a preset threshold $\gamma > 0$ w.r.t $A$ and $\texttt{rdf:type}$:

$$\text{EBR}(A) = \{x \in \Delta^\mathcal{I} \mid \phi(x, \texttt{rdf:type}, A) \geq \gamma\}. \tag{1}$$

- **Negation.** The EBR of the negation of a concept $C$ is the set of all entities in the domain $\Delta^\mathcal{I}$ excluding those in EBR($C$).

$$\text{EBR}(\neg C) = \Delta^\mathcal{I} \setminus \text{EBR}(C). \tag{2}$$

- **Conjunction/Disjunction.** The EBR of the conjunction/disjunction of concepts $C$ and $D$ is the intersection/union of their individual EBR's.

- **Existential restriction.** The EBR of an existential restriction $\exists\, u.C$ where $u \in \{r, r^{-1}\}$ consists of entities $x$ such that there exists an entity $y$ in EBR($C$) with a relation $u$ to $x$ scoring above $\gamma$.

$$\text{EBR}(\exists\, u.C) = \{x \mid \exists\, y : y \in \text{EBR}(C) \wedge \phi'(x, u, y) \geq \gamma\}, \tag{3}$$

where

$$\phi'(x, u, y) = \begin{cases} \phi(x, u, y) \text{ if } u = r \\ \phi(y, u, x) \text{ if } u = r^{-1} \end{cases} \tag{4}$$

- **Universal restriction.** Based on the fact that $\forall\, u.C = \neg(\exists\, u.\neg C)$, $\forall\, u \in \{r, r^{-1}\}$ we derive the EBR of universal restriction $\forall\, u.C$ as

$$\text{EBR}(\forall\, u.C) = \text{EBR}\big(\neg(\exists\, u.\neg C)\big). \tag{5}$$

- **Cardinality restriction.** The EBR of a cardinality restriction on a role $r$ and concept $C$ is the set of entities $x$ that have a number of $u$-related entities in EBR($C$) meeting the specified cardinality $\#n$, where $u \in \{r, r^{-1}\}$ and $\# \in \{\leq, \geq, =\}$.

$$\text{EBR}(\#n\, u.C) = \{x \ \mid |\{y|\phi'(x, u, y) \geq \gamma \wedge y \in \text{EBR}(C)\}|\#n\}\}. \tag{6}$$

with $\phi'$ defined in Equation 4.

Table 2: Syntax and neural semantics for nominals, top, bottom concepts and self restrictions.

| Concept Type | Syntax | Neural Semantics |
|---|---|---|
| Top concept | $\top$ | $\Delta^{\mathcal{I}}$ |
| Bottom concept | $\bot$ | $\emptyset$ |
| Nominals | $\{o_1, \ldots, o_n\}$ | $\{o_1, \ldots, o_n\}$ |
| Self-restriction | $\exists\, r.Self$ | $\{x : \phi(x, r, x) \geq \gamma\}$ |
| Inverse Self-restriction | $\exists\, r^{-1}.Self$ | $\{x : \phi(x, r, x) \geq \gamma\}$ |

Table 2 defines the neural semantics for nominals, top, and bottom concepts $\mathcal{SROIQ}$.

## 4 EXPERIMENTAL SETUP

### 4.1 DATASET

We evaluated our proposed approach on six benchmark datasets, including four large datasets: Carcinogenesis, Mutagenesis, Semantic Bible, and Vicodi, as well as two smaller datasets: Family and Father. These datasets cover a range of domains, from biological interactions to historical and familial relationships. Detailed statistics for each dataset are provided in the appendix (supplemental material).

### 4.2 EVALUATION

We evaluate our reasoner across three main tasks to assess its robustness and effectiveness.

In the first task, we focus on standard instance retrieval in a closed-world scenario using perfect knowledge basesspecifically, complete and consistent ones. The primary objective is to measure how effectively our reasoner retrieves instances from various datasets. To quantify this, we employ the Jaccard similarity and the F-measure, which compare instances retrieved by our reasoner ($\hat{y}$) to the ground truth ($y$). The Jaccard similarity $J$ as well as the F-measure $F_1$ are defined as:

$$J(\hat{y}, y) = \begin{cases} \dfrac{|\hat{y} \cap y|}{|\hat{y} \cup y|} & \text{if } y \neq \emptyset \text{ or } \hat{y} \neq \emptyset \\ 1 & \text{otherwise.} \end{cases} \qquad F_1(\hat{y}, y) = \begin{cases} 2 \times \dfrac{|\hat{y} \cap y|}{|\hat{y}| + |y|} & \text{if } y \neq \emptyset \text{ or } \hat{y} \neq \emptyset \\ 1 & \text{otherwise.} \end{cases} \qquad (7)$$

These metrics provide insight into how well our reasoner's predictions align with the true instances retrieved using a fast instance checker based on set-theoretic operations.

In the second set of experiments, we assess the performance of our reasoner when dealing with incomplete or noisy knowledge bases. Starting with a clean knowledge base, we introduce noise by adding false assertions or axioms at a specified level $\nu\%$. Additionally, we create incompleteness by removing a certain percentage ($\nu\%$) of axioms or assertions from the knowledge base. The goal is to evaluate our reasoner's ability to retrieve information with noisy and incomplete knowledge bases and to compare its performance with existing state-of-the-art methods.

## 5 RESULTS AND DISCUSSION

### 5.1 FIRST SET OF EXPERIMENTS: RETRIEVAL RESULTS IN A CLOSED WORD SCENARIO WITH COMPLETE AND CONSISTENT DATASETS

Table 3 shows the retrieval results of EBR in a closed-word scenario on full datasets. The results demonstrate that our reasoning approach achieves near-perfect retrieval performance in a closed-world scenario across all datasets, with consistently high Jaccard similarity and F1 scores. For named and negated concepts, as well as more complex constructs intersections and unions, existential and universal quantifications, and cardinality restrictions, the reasoner consistently returns

scores close to or equal to 1.000. This highlights the accuracy and robustness of EBR in retrieving $\mathcal{SROIQ}$ concept instances. Hence, this confirms that EBR is highly effective and reliable for concept retrieval in closed-world settings.

Table 3: Results of concept retrieval in a **closed-world** setting for all datasets. **#** denotes the number of concepts generated. The Jaccard similarity and the F1-score are computed. For cardinality restrictions, $n \in \{1, 2, 3\}$. For the set of named concepts $NC$ and negated named concepts $NNC$, we always choose $C$ and $D$ such that $C, D \in NC \cup NNC$.

| Concept | Syntax | Semantic Bible | | | Mutagenesis | | | Carcinogenesis | | |
|---|---|---|---|---|---|---|---|---|---|---|
| | | # | Jaccard | F1-score | # | Jaccard | F1-score | # | Jaccard | F1-score |
| named | $C$ | 48 | 1.000 | 1.000 | 86 | 0.999 | 0.999 | 142 | 1.000 | 1.000 |
| negated | $\neg C$ | 48 | 1.000 | 1.000 | 86 | 0.999 | 0.999 | 142 | 1.000 | 1.000 |
| intersection | $C \sqcap D$ | 576 | 1.000 | 1.000 | 289 | 0.999 | 0.999 | 196 | 1.000 | 1.000 |
| union | $C \sqcup D$ | 576 | 1.000 | 1.000 | 289 | 0.999 | 0.999 | 196 | 1.000 | 1.000 |
| existential | $\exists\, r.C$ | 180 | 1.000 | 1.000 | 68 | 1.000 | 1.000 | 56 | 1.000 | 1.000 |
| universal | $\forall\, r.C$ | 180 | 1.000 | 1.000 | 68 | 1.000 | 1.000 | 56 | 1.000 | 1.000 |
| min cardinality | $\geq nr.C$ | 96 | 1.000 | 1.000 | 96 | 1.000 | 1.000 | 4 | 1.000 | 1.000 |
| max cardinality | $\leq nr.C$ | 96 | 1.000 | 1.000 | 96 | 1.000 | 1.000 | 4 | 0.999 | 0.999 |
| exist nominals | $\exists\, r.\{o_1, \ldots, o_n\}$ | 480 | 1.000 | 1.000 | 240 | 1.000 | 1.000 | 240 | 1.000 | 1.000 |
| **Concept** | **Syntax** | **Vicodi** | | | **Father** | | | **Family** | | |
| | | # | Jaccard | F1-score | # | Jaccard | F1-score | # | Jaccard | F1-score |
| named | $C$ | 9 | 1.000 | 1.000 | 3 | 1.000 | 1.000 | 36 | 1.000 | 1.000 |
| negated | $\neg C$ | 9 | 1.000 | 1.000 | 3 | 1.000 | 1.000 | 36 | 1.000 | 1.000 |
| intersection | $C \sqcap D$ | 45 | 0.999 | 0.999 | 45 | 1.000 | 1.000 | 1620 | 1.000 | 1.000 |
| union | $C \sqcup D$ | 45 | 0.999 | 0.999 | 45 | 1.000 | 1.000 | 1620 | 1.000 | 1.000 |
| existential | $\exists\, r.C$ | 4 | 1.000 | 1.000 | 12 | 1.000 | 1.000 | 288 | 1.000 | 1.000 |
| universal | $\forall\, r.C$ | 4 | 0.999 | 0.999 | 12 | 1.000 | 1.000 | 288 | 1.000 | 1.000 |
| min cardinality | $\geq nr.C$ | 4 | 1.000 | 1.000 | 36 | 1.000 | 1.000 | 864 | 1.000 | 1.000 |
| max cardinality | $\leq nr.C$ | 4 | 0.999 | 0.999 | 36 | 1.000 | 1.000 | 864 | 1.000 | 1.000 |
| exist nominals | $\exists\, r.\{o_1, \ldots, o_n\}$ | 2 | 1.000 | 1.000 | 2 | 1.000 | 1.000 | 12 | 1.000 | 1.000 |

## 5.2 SECOND SET OF EXPERIMENTS: RETRIEVAL RESULTS IN A CLOSED WORLD SCENARIO WITH INCOMPLETE DATASETS

In Table 4, we present the results of instance retrieval under a closed-world scenario on incomplete datasets. The datasets were made incomplete at two levels: 40% incompleteness in the upper part of the table and 80% in the lower part (additional levels of incompleteness 10%, 20%, 60%, and 90% are provided in the appendix). For each dataset, five incomplete data samples were generated for the 40% and 80% incompleteness levels. The performance was computed for each sample, and the results were averaged, resulting in a total of 10 runs for the KGE evaluation at each incompleteness level (40% and 80%) for each dataset.

The comparison of results between EBR and symbolic methods like HermiT, Pellet, JFact, and Openllet highlights its superior performance in Jaccard similarity and runtime. In nearly every dataset, EBR achieves significantly higher Jaccard scores, indicating better accuracy in retrieval performance. For example, in the Family dataset, the EBR consistently outperforms others on complex concept types like OWLObjectAllValuesFrom, where it scores a Jaccard similarity of 0.528 compared to 0.000 by other methods. Similarly, for OWLObjectComplementOf, EBR leads with a 0.623 score, whereas the competing methods remain stagnant at 0.056. Additionally, in the Mutagenesis dataset, EBR also shows remarkable improvements, achieving a Jaccard similarity of 0.906 for OWLObjectIntersectionOf and 0.762 for OWLObjectUnionOf, far surpassing its counterparts.

Moreover, EBR offers considerable improvements in runtime efficiency. Across various datasets, it consistently records lower runtimes than the other approaches, particularly in handling complex concepts with large datasets. For example, in the Carcinogenesis dataset at 40% incompleteness, EBR completes the OWLObjectIntersectionOf query in 0.260 seconds. In contrast, other methods such as Pellet and JFact take significantly longer, with times as high as 6.469 seconds. This trend is further evident in the Mutagenesis dataset, where EBR reduces runtime to 0.368 seconds for OWLClass and 0.358 seconds on negated classes in contrast to JFact's 10.002 seconds and 9.762 seconds. This suggests that EBR significantly improves the computational efficiency, making it more suitable for large-scale, incomplete data scenarios.

### 5.3 Third Set of Experiments: Retrieval results in a closed world scenario with noisy datasets

In Table 5, we present the results of instance retrieval under a closed-world scenario on noisy datasets. For each dataset, we made them noisy by corrupting statements in the KB and adding them back at level 10% (Upper part of the Table) and 20% (Lower part of the Table). For each dataset in the Table, three samples of incomplete data were generated for both the 10% and 20% noise levels. The performance was computed for each sample, and the results were averaged, resulting in a total of 6 runs for the KGE evaluation at each incompleteness level (40% and 80%) for each datasets.

In the Table, we observe that standard reasoners failed to retrieve any instances in certain cases (indicated by dashes), as they marked the KB as inconsistent. This suggests that these reasoners struggle to handle noisy data. In the 10% noise scenario, EBR outperformed other reasoners across most concept types, especially for complex OWL expressions. For example, it achieved a Jaccard similarity of 0.986 for the OWLObjectMaxCardinality expression in the Family dataset, while all other reasoners failed to retrieve any instances and returned empty sets giving a Jaccard similarity of 0.000. We can observe similar performance on the Mutagenesis dataset, EBR delivered the highest Jaccard similarity (0.992) for OWLObjectComplementOf, with a much faster runtime than its counterparts.

## 6 Conclusion:

We introduced EBR, an embedding-based reasoner that leverages link prediction on knowledge graph embeddings to perform robust reasoning on noisy and incomplete DL KBs. Our experiments demonstrate that EBR significantly outperforms traditional symbolic reasoners, such as HermiT, Pellet, JFact, and Openllet, which often failed or declared the knowledge base inconsistent when faced with high levels of incompleteness or noise. In contrast, EBR maintained strong retrieval performance, even with up to 80% incompleteness, and consistently achieved high Jaccard similarity in noisy datasets with 10% and 20% noise levels. This resilience is due to EBR's ability to model relationships using embeddings, making it less sensitive to missing or inconsistent data, unlike symbolic reasoners that require complete and consistent datasets. This proves the claim that EBR is a scalable and effective solution for reasoning on real-world knowledge bases, where data imperfections are common.

Table 4: Retrieval performance on the datasets with $40\%$ incompleteness. **#** represents the number of expression types generated, **Jac** and **RT** represent the average Jaccard similarity and average runtime in seconds on every concept type. Bold values indicate that a particular approach outperforms others.

| Incomplete Datasets at 40% | Concept type | # | HermiT | | Pellet | | JFact | | Openllet | | EBR | |
|---|---|---|---|---|---|---|---|---|---|---|---|---|
| | | | Jac | RT | Jac | RT | Jac | RT | Jac | RT | Jac | RT |
| **Father** | OWLClass | 3 | 0.639 | 0.017 | 0.639 | 0.002 | 0.639 | 0.002 | 0.639 | 0.002 | 0.639 | 0.001 |
| | OWLObjectAllValuesFrom | 12 | 0.111 | 0.009 | 0.111 | 0.006 | 0.111 | 0.003 | 0.111 | 0.003 | **0.544** | 0.001 |
| | OWLObjectComplementOf | 3 | 0.750 | 0.003 | 0.750 | 0.002 | 0.750 | 0.002 | 0.750 | 0.002 | 0.750 | 0.001 |
| | OWLObjectIntersectionOf | 36 | 0.824 | 0.030 | 0.774 | 0.008 | 0.774 | 0.004 | 0.774 | 0.002 | 0.824 | 0.001 |
| | OWLObjectMaxCardinality | 36 | 0.111 | 0.007 | 0.111 | 0.003 | 0.111 | 0.003 | 0.111 | 0.003 | **0.670** | 0.001 |
| | OWLObjectMinCardinality | 36 | 0.838 | 0.009 | 0.838 | 0.004 | 0.838 | 0.005 | 0.838 | 0.002 | 0.838 | 0.001 |
| | OWLObjectSomeValuesFrom | 12 | 0.597 | 0.004 | 0.597 | 0.002 | 0.597 | 0.003 | 0.597 | 0.013 | 0.597 | 0.001 |
| | OWLObjectUnionOf | 36 | 0.657 | 0.004 | 0.657 | 0.002 | 0.657 | 0.003 | 0.657 | 0.005 | 0.657 | 0.001 |
| **Family** | OWLClass | 18 | 0.613 | 0.012 | 0.613 | 0.006 | 0.613 | 0.014 | 0.613 | 0.006 | 0.613 | 0.005 |
| | OWLObjectAllValuesFrom | 288 | 0.000 | 0.135 | 0.000 | 0.007 | 0.000 | 0.011 | 0.000 | 0.007 | **0.528** | 0.072 |
| | OWLObjectComplementOf | 18 | 0.056 | 0.141 | 0.056 | 0.006 | 0.056 | 0.008 | 0.056 | 0.006 | **0.623** | 0.005 |
| | OWLObjectIntersectionOf | 1296 | 0.316 | 0.080 | 0.316 | 0.008 | 0.316 | 0.012 | 0.316 | 0.007 | **0.687** | 0.010 |
| | OWLObjectMaxCardinality | 864 | 0.000 | 0.136 | 0.000 | 0.009 | 0.000 | 0.011 | 0.000 | 0.008 | **0.595** | 0.039 |
| | OWLObjectMinCardinality | 864 | 0.466 | 0.144 | 0.466 | 0.011 | 0.466 | 0.013 | 0.466 | 0.010 | **0.614** | 0.039 |
| | OWLObjectSomeValuesFrom | 288 | 0.260 | 0.137 | 0.260 | 0.008 | 0.260 | 0.011 | 0.260 | 0.008 | **0.442** | 0.039 |
| | OWLObjectUnionOf | 1296 | 0.318 | 0.108 | 0.318 | 0.009 | 0.318 | 0.012 | 0.318 | 0.008 | **0.606** | 0.010 |
| **Semantic Bible** | OWLClass | 5 | 0.337 | 0.072 | 0.337 | 0.018 | 0.337 | 0.049 | 0.337 | 0.018 | 0.337 | 0.528 |
| | OWLObjectComplementOf | 5 | 0.013 | 1.750 | 0.013 | 0.053 | 0.013 | 0.032 | 0.013 | 0.040 | **0.273** | 2.678 |
| | OWLObjectIntersectionOf | 20 | 0.587 | 1.465 | 0.587 | 0.040 | 0.587 | 0.085 | 0.587 | 0.020 | **0.653** | 2.468 |
| | OWLObjectUnionOf | 20 | 0.226 | 0.471 | 0.226 | 0.030 | 0.226 | 0.057 | 0.226 | 0.020 | **0.291** | 1.093 |
| **Mutagenesis** | OWLClass | 5 | **0.915** | 0.465 | **0.915** | 0.455 | **0.915** | 10.002 | **0.915** | 0.456 | 0.914 | 0.368 |
| | OWLObjectComplementOf | 5 | 0.000 | 373.715 | 0.000 | 0.465 | 0.000 | 9.762 | 0.000 | 0.443 | **0.711** | 0.358 |
| | OWLObjectIntersectionOf | 20 | 0.729 | 281.808 | 0.729 | 0.478 | 0.729 | 9.937 | 0.729 | 0.448 | **0.906** | 5.408 |
| | OWLObjectUnionOf | 20 | 0.597 | 94.854 | 0.597 | 0.567 | 0.597 | 9.756 | 0.597 | 0.563 | **0.762** | 0.814 |
| **Carcinogenesis** | OWLClass | 4 | 0.119 | 0.086 | 0.119 | 0.076 | 0.119 | 0.966 | 0.119 | 0.094 | 0.119 | 0.205 |
| | OWLObjectComplementOf | 4 | 0.099 | 3.389 | 0.099 | 0.143 | 0.099 | 1.117 | 0.099 | 0.117 | **0.120** | 4.255 |
| | OWLObjectIntersectionOf | 32 | **0.419** | 6.469 | **0.419** | 0.107 | **0.419** | 1.120 | **0.419** | 0.091 | 0.387 | 0.260 |
| | OWLObjectUnionOf | 32 | 0.111 | 2.227 | 0.111 | 0.117 | 0.111 | 1.081 | 0.111 | 0.135 | **0.121** | 0.791 |
| **Vicodi** | OWLClass | 15 | 0.195 | 0.168 | 0.195 | 0.081 | 0.195 | 1.208 | 0.195 | 0.079 | 0.195 | 2.166 |
| | OWLObjectComplementOf | 15 | 0.047 | 4.810 | 0.047 | 0.087 | 0.047 | 1.332 | 0.047 | 0.085 | **0.050** | 3.097 |
| | OWLObjectIntersectionOf | 180 | 0.424 | 14.618 | 0.424 | 0.093 | 0.424 | 1.246 | 0.424 | 0.086 | **0.427** | 4.724 |
| | OWLObjectUnionOf | 180 | 0.076 | 4.428 | 0.076 | 0.103 | 0.076 | 1.227 | 0.076 | 0.102 | **0.077** | 5.488 |
| **Incomplete Datasets at 80%** | **Concept Type** | **#** | **HermiT** | | **Pellet** | | **JFact** | | **Openllet** | | **EBR** | |
| | | | Jac | RT | Jac | RT | Jac | RT | Jac | RT | Jac | RT |
| **Father** | OWLClass | 3 | 0.278 | 0.002 | 0.278 | 0.002 | 0.278 | 0.002 | 0.278 | 0.002 | **0.639** | 0.001 |
| | OWLObjectAllValuesFrom | 12 | 0.056 | 0.008 | 0.056 | 0.002 | 0.056 | 0.003 | 0.056 | 0.002 | **0.544** | 0.001 |
| | OWLObjectComplementOf | 3 | 0.500 | 0.002 | 0.500 | 0.002 | 0.500 | 0.002 | 0.500 | 0.002 | **0.750** | 0.001 |
| | OWLObjectIntersectionOf | 36 | 0.648 | 0.030 | 0.648 | 0.006 | 0.648 | 0.004 | 0.648 | 0.002 | **0.824** | 0.001 |
| | OWLObjectMaxCardinality | 36 | 0.056 | 0.005 | 0.056 | 0.002 | 0.056 | 0.003 | 0.056 | 0.002 | **0.670** | 0.001 |
| | OWLObjectMinCardinality | 36 | 0.694 | 0.008 | 0.694 | 0.002 | 0.694 | 0.003 | 0.694 | 0.002 | **0.838** | 0.001 |
| | OWLObjectSomeValuesFrom | 12 | 0.167 | 0.007 | 0.167 | 0.002 | 0.167 | 0.003 | 0.167 | 0.002 | **0.597** | 0.001 |
| | OWLObjectUnionOf | 36 | 0.315 | 0.006 | 0.315 | 0.003 | 0.315 | 0.003 | 0.315 | 0.005 | **0.657** | 0.001 |
| **Family** | OWLClass | 18 | 0.210 | 0.003 | 0.210 | 0.003 | 0.210 | 0.004 | 0.210 | 0.003 | 0.210 | 0.003 |
| | OWLObjectAllValuesFrom | 288 | 0.000 | 0.027 | 0.000 | 0.004 | 0.000 | 0.006 | 0.000 | 0.003 | **0.173** | 0.020 |
| | OWLObjectComplementOf | 18 | 0.056 | 0.022 | 0.056 | 0.003 | 0.056 | 0.004 | 0.056 | 0.003 | **0.241** | 0.004 |
| | OWLObjectIntersectionOf | 1296 | 0.239 | 0.017 | 0.239 | 0.003 | 0.239 | 0.006 | 0.239 | 0.003 | **0.361** | 0.007 |
| | OWLObjectMaxCardinality | 864 | 0.000 | 0.030 | 0.000 | 0.005 | 0.000 | 0.005 | 0.000 | 0.003 | **0.202** | 0.012 |
| | OWLObjectMinCardinality | 864 | 0.406 | 0.027 | 0.406 | 0.004 | 0.406 | 0.005 | 0.406 | 0.003 | **0.413** | 0.012 |
| | OWLObjectSomeValuesFrom | 288 | 0.079 | 0.025 | 0.079 | 0.003 | 0.079 | 0.006 | 0.079 | 0.003 | **0.099** | 0.012 |
| | OWLObjectUnionOf | 1296 | 0.109 | 0.020 | 0.109 | 0.004 | 0.109 | 0.005 | 0.109 | 0.003 | **0.204** | 0.007 |
| **Semantic Bible** | OWLClass | 5 | 0.100 | 0.014 | 0.100 | 0.012 | 0.100 | 0.030 | 0.100 | 0.010 | 0.100 | 0.335 |
| | OWLObjectComplementOf | 5 | 0.000 | 0.200 | 0.000 | 0.011 | 0.000 | 0.022 | 0.000 | 0.012 | **0.051** | 0.329 |
| | OWLObjectIntersectionOf | 20 | 0.525 | 0.153 | 0.525 | 0.034 | 0.525 | 0.028 | 0.525 | 0.028 | **0.538** | 1.462 |
| | OWLObjectUnionOf | 20 | 0.050 | 0.203 | 0.050 | 0.059 | 0.050 | 0.034 | 0.050 | 0.016 | **0.063** | 0.696 |
| **Mutagenesis** | OWLClass | 5 | 0.200 | 0.426 | 0.200 | 0.050 | 0.200 | 0.223 | 0.200 | 0.038 | 0.200 | 0.124 |
| | OWLObjectComplementOf | 5 | 0.000 | 3.332 | 0.000 | 0.127 | 0.000 | 0.243 | 0.000 | 0.046 | **0.050** | 0.643 |
| | OWLObjectIntersectionOf | 20 | 0.550 | 2.267 | 0.550 | 0.039 | 0.550 | 0.223 | 0.550 | 0.039 | **0.563** | 1.663 |
| | OWLObjectUnionOf | 20 | 0.075 | 0.791 | 0.075 | 0.042 | 0.075 | 0.195 | 0.075 | 0.029 | **0.088** | 0.296 |
| **Carcinogenesis** | OWLClass | 4 | 0.119 | 0.086 | 0.119 | 0.076 | 0.119 | 0.966 | 0.119 | 0.094 | 0.119 | 0.205 |
| | OWLObjectComplementOf | 4 | 0.099 | 3.389 | 0.099 | 0.143 | 0.099 | 1.117 | 0.099 | 0.117 | **0.120** | 4.255 |
| | OWLObjectIntersectionOf | 32 | **0.419** | 6.469 | **0.419** | 0.107 | **0.419** | 1.120 | **0.419** | 0.091 | 0.387 | 0.260 |
| | OWLObjectUnionOf | 32 | 0.111 | 2.227 | 0.111 | 0.117 | 0.111 | 1.081 | 0.111 | 0.135 | **0.121** | 0.791 |
| **Vicodi** | OWLClass | 15 | 0.195 | 0.168 | 0.195 | 0.081 | 0.195 | 1.208 | 0.195 | 0.079 | 0.195 | 2.166 |
| | OWLObjectComplementOf | 15 | 0.047 | 4.810 | 0.047 | 0.087 | 0.047 | 1.332 | 0.047 | 0.085 | **0.050** | 3.097 |
| | OWLObjectIntersectionOf | 180 | 0.424 | 14.618 | 0.424 | 0.093 | 0.424 | 1.246 | 0.424 | 0.086 | **0.427** | 4.724 |
| | OWLObjectUnionOf | 180 | 0.076 | 4.428 | 0.076 | 0.103 | 0.076 | 1.227 | 0.076 | 0.102 | **0.077** | 5.488 |

Table 5: Retrieval performance on noisy datasets. **#** represents the number of expression types generated, **Jac** and **RT** represent the average Jaccard similarity and average runtime in seconds on every concept type. The dash (-) means that the reasoners failed to retrieve any instances.

| Noisy Datasets at 10% / Concept Type | # | HermiT | | Pellet | | JFact | | Openllet | | EBR | |
|---|---|---|---|---|---|---|---|---|---|---|---|
| | | Jac | RT | Jac | RT | Jac | RT | Jac | RT | Jac | RT |
| **Father** | | | | | | | | | | | |
| OWLClass | 9 | - | - | - | - | - | - | - | - | 0.852 | 0.001 |
| OWLObjectAllValuesFrom | 36 | - | - | - | - | - | - | - | - | 0.781 | 0.002 |
| OWLObjectComplementOf | 9 | - | - | - | - | - | - | - | - | 0.861 | 0.001 |
| OWLObjectIntersectionOf | 108 | - | - | - | - | - | - | - | - | 0.850 | 0.002 |
| OWLObjectMaxCardinality | 108 | - | - | - | - | - | - | - | - | 0.963 | 0.002 |
| OWLObjectMinCardinality | 108 | - | - | - | - | - | - | - | - | 0.749 | 0.002 |
| OWLObjectSomeValuesFrom | 36 | - | - | - | - | - | - | - | - | 0.711 | 0.002 |
| OWLObjectUnionOf | 108 | - | - | - | - | - | - | - | - | 0.888 | 0.002 |
| **Family** | | | | | | | | | | | |
| OWLClass | 18 | 0.879 | 0.004 | 0.879 | 0.002 | 0.879 | 0.010 | 0.879 | 0.002 | 0.879 | 0.008 |
| OWLObjectAllValuesFrom | 288 | 0.000 | 0.399 | 0.000 | 0.001 | 0.000 | 0.010 | 0.000 | 0.001 | **0.906** | 0.244 |
| OWLObjectComplementOf | 18 | 0.056 | 0.395 | 0.056 | 0.003 | 0.056 | 0.014 | 0.056 | 0.004 | **0.918** | 0.010 |
| OWLObjectIntersectionOf | 1296 | 0.298 | 0.218 | 0.298 | 0.001 | 0.298 | 0.011 | 0.298 | 0.001 | **0.795** | 0.021 |
| OWLObjectMaxCardinality | 864 | 0.000 | 0.400 | 0.000 | 0.001 | 0.000 | 0.010 | 0.000 | 0.001 | **0.986** | 0.124 |
| OWLObjectMinCardinality | 864 | 0.536 | 0.425 | 0.536 | 0.010 | 0.536 | 0.011 | 0.536 | 0.011 | **0.671** | 0.124 |
| OWLObjectSomeValuesFrom | 288 | 0.469 | 0.405 | 0.469 | 0.002 | 0.469 | 0.012 | 0.469 | 0.002 | **0.817** | 0.124 |
| OWLObjectUnionOf | 1296 | 0.501 | 0.306 | 0.501 | 0.002 | 0.501 | 0.011 | 0.501 | 0.002 | **0.934** | 0.020 |
| **Semantic Bible** | | | | | | | | | | | |
| OWLClass | 2 | - | - | - | - | - | - | - | - | 0.637 | 1.069 |
| OWLObjectComplementOf | 2 | - | - | - | - | - | - | - | - | 0.854 | 1.023 |
| OWLObjectIntersectionOf | 8 | - | - | - | - | - | - | - | - | 0.873 | 2.210 |
| OWLObjectUnionOf | 8 | - | - | - | - | - | - | - | - | 0.812 | 2.114 |
| **Carcinogenesis** | | | | | | | | | | | |
| OWLClass | 4 | - | - | - | - | - | - | - | - | 0.553 | 1.443 |
| OWLObjectComplementOf | 4 | - | - | - | - | - | - | - | - | 0.999 | 1.396 |
| OWLObjectIntersectionOf | 32 | - | - | - | - | - | - | - | - | 0.710 | 3.309 |
| OWLObjectUnionOf | 32 | - | - | - | - | - | - | - | - | 0.922 | 3.420 |
| **Mutagenesis** | | | | | | | | | | | |
| OWLClass | 2 | **0.924** | 0.330 | **0.924** | 0.149 | **0.924** | 25.699 | **0.924** | 0.144 | 0.913 | 0.095 |
| OWLObjectComplementOf | 2 | 0.000 | 1709.678 | 0.000 | 0.151 | 0.000 | 28.742 | 0.000 | 0.158 | **0.992** | 0.107 |
| OWLObjectIntersectionOf | 8 | 0.731 | 1179.387 | 0.731 | 0.193 | 0.731 | 24.844 | 0.731 | 0.118 | **0.976** | 1.776 |
| OWLObjectUnionOf | 8 | 0.731 | 425.885 | 0.731 | 0.317 | 0.731 | 25.424 | 0.731 | 0.273 | **0.973** | 0.269 |
| **Noisy Datasets at 20% / Concept Type** | # | HermiT | | Pellet | | JFact | | Openllet | | EBR | |
| | | Jac | RT | Jac | RT | Jac | RT | Jac | RT | Jac | RT |
| **Father** | | | | | | | | | | | |
| OWLClass | 9 | - | - | - | - | - | - | - | - | 0.744 | 0.001 |
| OWLObjectAllValuesFrom | 36 | - | - | - | - | - | - | - | - | 0.647 | 0.002 |
| OWLObjectComplementOf | 9 | - | - | - | - | - | - | - | - | 0.583 | 0.001 |
| OWLObjectIntersectionOf | 108 | - | - | - | - | - | - | - | - | 0.707 | 0.002 |
| OWLObjectMaxCardinality | 108 | - | - | - | - | - | - | - | - | 0.887 | 0.002 |
| OWLObjectMinCardinality | 108 | - | - | - | - | - | - | - | - | 0.595 | 0.002 |
| OWLObjectSomeValuesFrom | 36 | - | - | - | - | - | - | - | - | 0.562 | 0.002 |
| OWLObjectUnionOf | 108 | - | - | - | - | - | - | - | - | 0.752 | 0.002 |
| **Family** | | | | | | | | | | | |
| OWLClass | 18 | 0.780 | 0.005 | 0.780 | 0.002 | 0.780 | 0.007 | 0.780 | 0.002 | 0.780 | 0.006 |
| OWLObjectAllValuesFrom | 288 | 0.000 | 0.496 | 0.000 | 0.001 | 0.000 | 0.013 | 0.000 | 0.001 | **0.817** | 0.197 |
| OWLObjectComplementOf | 18 | 0.056 | 0.495 | 0.056 | 0.001 | 0.056 | 0.009 | 0.056 | 0.001 | **0.840** | 0.007 |
| OWLObjectIntersectionOf | 1296 | 0.268 | 0.272 | 0.268 | 0.001 | 0.268 | 0.011 | 0.268 | 0.001 | **0.680** | 0.017 |
| OWLObjectMaxCardinality | 864 | 0.000 | 0.499 | 0.000 | 0.001 | 0.000 | 0.010 | 0.000 | 0.001 | **0.973** | 0.109 |
| OWLObjectMinCardinality | 864 | 0.514 | 0.536 | 0.514 | 0.012 | 0.514 | 0.012 | 0.514 | 0.013 | **0.538** | 0.111 |
| OWLObjectSomeValuesFrom | 288 | 0.404 | 0.511 | 0.404 | 0.002 | 0.404 | 0.012 | 0.404 | 0.003 | **0.699** | 0.111 |
| OWLObjectUnionOf | 1296 | 0.489 | 0.381 | 0.489 | 0.002 | 0.489 | 0.012 | 0.489 | 0.002 | **0.878** | 0.018 |
| **Semantic Bible** | | | | | | | | | | | |
| OWLClass | 2 | - | - | - | - | - | - | - | - | 0.205 | 7.519 |
| OWLObjectComplementOf | 2 | - | - | - | - | - | - | - | - | 0.783 | 7.254 |
| OWLObjectIntersectionOf | 8 | - | - | - | - | - | - | - | - | 0.747 | 14.649 |
| OWLObjectUnionOf | 8 | - | - | - | - | - | - | - | - | 0.692 | 14.387 |

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
