# OpenReview forum: "Neural Description Logic Reasoning over Incomplete Knowledge Bases"
_ICLR.cc/2025/Conference — Submitted to ICLR 2025_

### Official Review · Reviewer_jXrQ · 2024-10-22

**Soundness:** 3
**Presentation:** 3
**Contribution:** 2
**Rating:** 3
**Confidence:** 2

**Summary:**

This paper proposes a novel neural reasoner, dubbed EBR, reasoning over incomplete and inconsistent knowledge graphs. Authors propose a neural interpretation for the SROIQ semantics for Descriptive Logic. A substantial survey is conducted. Experiments are carried out in six datasets, with very good results -- achieving near-perfect results in the close world scenario.

**Strengths:**

This paper addresses an important issue of knowledge graph reasoning, and follows the embedding approach to deal with the incompleteness and inconsistency of knowledge graphs. The paper is very well polished, from writing to experiments.

**Weaknesses:**

It is not clear what neural architectures are used. The datasets used in the experiments are not those used by SOTA papers. After inspecting the supplementary material, I see codes, but do not find datasets. Authors described in section 2.3 that CQD and other neural logical query methods do not support negation, universal restriction and cardinality restriction. However, Beta-E supports:

Beta Embeddings for Multi-Hop Logical Reasoning in Knowledge Graphs. H. Ren, J. Leskovec. Neural Information Processing Systems (NeurIPS), 2020.

And CQD defines the complementary t-conorm as \bottom (x,y) = 1 - \top(1-x, 1-y). This automatically follows ways of definitions of negation. If \top(x,y) = min(x,y), then, -x = -\top(x,x) = -\top(1- (1-x), 1- (1-x)) = \bottom(1-x, 1-x) -1.

In Table 2, the neural semantics of \Delta^\mathcal{I} and \emptyset are the same as the semantics of those in Table 1.

**Questions:**

1. are the datasets publicly available?

2. line 210: "mapping of DL syntax to a neural semantic syntax". You are mapping " .. syntax to .. syntax", or ".. syntax to a neural semantics"?

3. what the neural architectures of the proposed method?

4. In section 3, what is the novelty in methodology?

---

### Official Review · Reviewer_UN7H · 2024-10-26

**Soundness:** 3
**Presentation:** 2
**Contribution:** 1
**Rating:** 3
**Confidence:** 4

**Summary:**

The paper claims to present a neural reasoner that captures the full semantics of the description logic SROIQ while scaling to large knoweldge bases. The paper discusses related work, presents the reasoning approach and evaluates the correctness of the model in three different closed world scenarios.

**Strengths:**

Combining symbolic reasoning with neural methods is a very promising approach to mitigate the known problems of the two approaches, containing the ones mentioned in the paper.

**Weaknesses:**

The approach proposed is quite naive in my opinion: use an existing link prediction approach to answer triple queries and combine the results according to the semantics of description logics operators. This approach has a number of limitations and does not really preserve the logical semantics of SROIQ:

- the operators defined in the paper seem to differ from the one's I would expect in SROIQ. for instance I missed how complex role inclusions are handled. Further, I was surprised to see the 'self' operator in the definition, also this typically makes expressive description logics undecidable
- The reasoning seems to rely on a the closed-world assumption (the evalution uses a closed word setting), which is not the semantics of SROIQ that uses open World semantics and features real negation as well as ways to implicitly formulate negation.

The evaluation - as mentioned above - is not really suited to show that the approach preserves SROIQ semantics. The setting is much closer to complex querying over a database than logical reasoning. Given this, I miss a comparison with more database-like approach to neural symbolic reasoning, e.g. based on datalog queries.

**Questions:**

- How are complex role assertions handled by your reasoner?
- How can you deal with the open world semantics underlying SROIQ?
- What other approach to combining complex database queries with neural approach exist and how does your approach perform in comparison to these?

---

### Official Review · Reviewer_JEta · 2024-11-04

**Soundness:** 2
**Presentation:** 2
**Contribution:** 2
**Rating:** 5
**Confidence:** 2

**Summary:**

This paper proposes a novel embedded reasoning model called Embedding Based Reasoner (EBR), aimed at addressing the issues of incompleteness and inconsistency in the Knowledge Base (KB). The traditional symbolic inference engine is inefficient and not robust enough when dealing with large-scale or erroneous data KB. In this paper, the neural inference engine EBR overcomes these shortcomings by quickly approximating the inference results of the symbolic inference engine through embedding technology.

**Strengths:**

* The paper is well organized.
* Experiments carried out by authors are sufficient.

**Weaknesses:**

* The theoretical explanation of the method is limited: EBR uses embedded reasoning techniques, but there is insufficient detailed explanation of its theoretical basis and working principle. Lack of in-depth analysis of the consistency and interpretability of embedded models in DL semantics may affect trust in the robustness and reliability of the method.
* Although EBR has significantly improved efficiency on large-scale datasets, there is a lack of detailed quantitative analysis of its computational resource requirements, such as memory consumption and GPU computing resources.

**Questions:**

* Is there a critical point or noise level that significantly reduces the performance of EBR? Can you provide some applicability conditions?
* How does EBR ensure consistency between embedded representations and symbolic inference logic?

---

### Official Review · Reviewer_kAv9 · 2024-11-05

**Soundness:** 2
**Presentation:** 2
**Contribution:** 1
**Rating:** 3
**Confidence:** 4

**Summary:**

This paper presents Embedding-Based Reasoner (EBR), a neural description logic reasoner designed to handle large-scale, incomplete, and inconsistent knowledge bases. EBR approximates logical reasoning under the $\mathcal{SHOIQ}$ syntax by using existing neural embeddings for the KB, aiming to provide a scalable and robust solution for handling noisy data. The experiment results demonstrate superior instance retrieval performance of EBR over conventional symbolic reasoners including HermiT, Pellet, JFact, and Openllet, across several benchmark datasets.

**Strengths:**

(1) This work explores the important field of neuro-symbolic reasoning, which is crucial for advancing knowledge representation and reasoning, especially for real-world applications where incomplete or noisy data is unavoidable.

(2) The time efficiency for performing reasoning on large, noisy KBs is also important in practice.

(3) The paper provides detailed background of description logic and SHOIQ syntax, offering clear formulations that help readers understand the context of the task and the proposed approach.

**Weaknesses:**

(1) First, the technical contribution is unclear. Although the paper introduces EBR as a novel neural reasoner for incomplete/inconsistent KBs, it heavily relies on existing neural embedding techniques, with limited originality beyond adopting the embeddings for DL-based reasoning. This makes it unclear what aspects of EBR are technically new.

I understand the theoretical contribution as to introduce the mapping between DL syntax and neural semantics. However, as EBR is only applied to the task of instance retrieval, it remains unclear that, whether, and to what extent, the mappings help to improve the performance of the reasoner. Besides, no evidence (e.g., provable guarantee) was given to justify the correctness on the theoretical side, which further limits the significance of the work.

Several improvements could be done to improve the paper, including (i) present explicit comparison about how EBR fundamentally differs from prior approaches; (ii) provide evidence (if any) such as theoretical guarantee to validate the contribution.

(2) The evaluation process is unclear. Section 4 and Section 5 do not explain in detail how to conduct “instance retrieval” in a given KB. How does the EBR reasoner work in the experiments? For example, does this process involve reasoning over graph structure? How to compute the score for each entity with each concept?

Section 3.2 introduces the link prediction task, which is a standard task over graph data, However, the experiments only conduct instance retrieval but not link prediction. Does instance retrieval relate to link prediction? If so, this should be clarified to avoid confusion. If not, then what is the purpose to mention link prediction in Section 3.2?

How were the KB embeddings trained? Does the KB embedding training process form a part of EBR? All the details including input/output scheme, encoding/decoding process, message passing mechanism, loss function, etc., are essential for readers to understand the working process and the utility of EBR. (The detailed settings might be given in the appendixes, but the current version does not contain them.)

(3) Lack of Comparisons with Neural Embedding-based Models. In the paper, EBR is only compared with traditional symbolic reasoners, while there is no comparison with recent neural-based or hybrid models that can also handle incomplete data (e.g., rule learning models including Neural-LP[1], DRUM[2], or ontology-aware neural models).

[1] Fan Yang, Zhilin Yang, William W. Cohen. Differentiable Learning of Logical Rules for Knowledge Base Reasoning. NeurIPS 2017

[2] Ali Sadeghian, Mohammadreza Armandpour, Patrick Ding, Daisy Zhe Wang. DRUM: End-To-End Differentiable Rule Mining On Knowledge Graphs. NeurIPS 2019

(4) Case study and detailed analysis should be presented. The current evaluation only reports the Jaccard similarity, F1 scores and running time for instance retrieval, which are all high-level statistics and provide little insights about the underlying work process and benefits of EBR.

To improve this, instead of simply reporting the metric scores on every datasets, I suggest the authors to include analysis of some cases extracted from any dataset. For example, by comparing the different performance of EBR and the baselines, readers could gain more insights about why EBR or any baseline makes it correct/incorrect.

**Minor issues**

(1) Line 101, “…iff $C^\mathcal{I} \sqsubseteq D^\mathcal{I}$…” should be “$C^\mathcal{I} \subseteq D^\mathcal{I}$”.

(2) Line 241, “The syntax and semantics for concepts in SROIQ are provided in the appendix.”---They are not in the appendix.

(3) All the tables in the appendix need to be discussed. Leaving the tables alone without any analysis provides little information for the readers.

**Questions:**

(1) Section 3.2 introduces the link prediction task, which is a standard task over graph data, but how does it relate to instance retrieval that is conducted in the experiments? Also, it seems no (standard) link prediction experiment was conducted. If so, what is the purpose to mention link prediction in Section 3.2?

(2) How were the KB embeddings trained? Does the KB embedding training process form a part of EBR? These details are essential for readers to understand the working process and utility of EBR.

(3) The authors claim that EBR could scale to large datasets. But according to the dataset statistics in the appendix, the largest dataset Vicodi only has 33K instances and 116K assertions. On the other hand, real-world KBs such as Freebase, DBpedia, are typically in million or even billion scale. I wonder if there are any standard convention to conceptualize “large-scale KBs"? I am also curious about whether the proposed EBR can handle KBs at the scale such as Freebase?

---

### Official Review · Reviewer_S1Dk · 2024-11-09

**Soundness:** 2
**Presentation:** 4
**Contribution:** 1
**Rating:** 3
**Confidence:** 5

**Summary:**

This is a very well-written paper on an interesting problem. The paper is mostly sound. But it downplays the capabilities of related work and overpromises on what it accomplishes. Furthermore, the originality of the proposed approach is minimal - which might perhaps be ok if there were a comprehensive evaluation, which, however, is not part of the paper.

**Strengths:**

1 The targeted problem is interesting
2 The proposed approach can be easily reproduced

**Weaknesses:**

1 The paper overpromises
2 Suggested method is a trivial extension of existing methods
3 Experimental comparison with related work is weak
4 Description of the experimental setup is weak


*1 The paper overpromises*
It says "We propose neural semantics to tackle the instance retrieval problem on incomplete or inconsistent SROIQ KBs".
What the paper actually does is that it (step A) computes a backbone based on a very limited set of axioms only containing instance assertions, role assertions, and subsumption axioms of the explicit form (C rdfs:subclassOf D). This is a tiny subset of SROIQ axioms.
Based on this subset, it allows for (step B) the querying of SROIQ concepts from a very limited set of queries, i.e., the ones listed in Table 3, but no recursive definition of concept expressions was applied, again underutilizing the capabilities of SROIQ (at least I could not read this from the paper).

*2 Suggested method is a trivial extension of existing methods*
The two steps (step A) and (step B) could have been trivially done by a range of Complex Query Answering methods.

*3 Experimental comparison with related work is weak*
The proposed approach is an approximation. The only comparisons are made against sound (and complete) semantic reasoners. Other trivially available approximations are not considered. As mentioned above, complex query-answering methods will be available. Even if few constructs would not be available in a particular answering method, others would be and could be compared.
Similarly, maybe worse, the statement that description logic embeddings do not support instance retrieval is wrong. Already, the computation of the backbone in these methods could have been more powerful than (step A) suggested here, and a comparison to their approximation would be easily possible. Note, for example, that a union query would be trivially available for box or ball embeddings by exactly returning the disjunction of the two elements. Note that this is a *trivial* modification and does not change these suggested methods, since no complex composition of concept expressions is required.

*4 Description of the experimental setup is weak*
The procedure for constructing wrong axioms is unspecified.
The procedure for having queries remains vague (are retrievals of composed concept expressions part of the queries?)
It is unclear to what extent the benchmark datasets do or do not exploit the expressiveness of SROIQ.

**Questions:**

Please clarify issues I mentioned under weaknesses, esp. ones related to *4 Description of the experimental setup is weak*

---

### Meta-Review · Area_Chair_yyHA · 2024-12-20

**Metareview:**

This paper presents Embedding-Based Reasoner (EBR), a neural description logic reasoner designed to handle large-scale, incomplete, and inconsistent knowledge bases. EBR approximates logical reasoning under the SHOIQ syntax by using existing neural embeddings for the KB, aiming to provide a scalable and robust solution for handling noisy data. Although the problem formulation of this work is both interesting and important, and the paper is well-written; however, it also has some flaws and gaps. The theoretical analysis in the paper is limited, and the experimental section also struggles to provide convincing results.
Specifically, the paper does not provide a comparison with state-of-the-art (SOTA) experimental results.  Thus, I recommend rejecting this paper.

**Additional Comments On Reviewer Discussion:**

The author of the paper did not provide a rebuttal.

---

### Decision · Program_Chairs · 2025-01-22

Reject